# Caffeine-Induced Acute and Delayed Responses in Cerebral Metabolism of Control and Schizophrenia-like Wisket Rats

**DOI:** 10.3390/ijms23158186

**Published:** 2022-07-25

**Authors:** Gyöngyi Horvath, István Kertész, Tamás Nagy, Leatitia Gabriella Adlan, Gabriella Kekesi, Alexandra Büki, Gabor Tuboly, György Trencsényi

**Affiliations:** 1Department of Physiology, Albert Szent-Györgyi Medical School, University of Szeged, H-6720 Szeged, Hungary; adlan.leatitia@med.u-szeged.hu (L.G.A.); kekesi.gabriella@med.u-szeged.hu (G.K.); buki.alexandra@med.u-szeged.hu (A.B.); 2Department of Medical Imaging, Division of Nuclear Medicine and Translational Imaging, Faculty of Medicine, University of Debrecen, H-4032 Debrecen, Hungary; kertesz.istvan@med.unideb.hu (I.K.); nagy.tamas@med.unideb.hu (T.N.); trencsenyi.gyorgy@med.unideb.hu (G.T.); 3Department of Neurology, Albert Szent-Györgyi Clinical Center, University of Szeged, H-6720 Szeged, Hungary; tuboly.gabor@med.u-szeged.hu

**Keywords:** brain metabolism, caffeine, multiple hit, PET, schizophrenia

## Abstract

Recently, morphological impairments have been detected in the brain of a triple-hit rat schizophrenia model (Wisket), and delayed depressive effects of caffeine treatment in both control and Wisket animals have also been shown. The aims of this study were to determine the basal and caffeine-induced acute (30 min) and delayed (24 h) changes in the cerebral ^18^fluorodeoxyglucose (^18^F-FDG) uptake by positron emission tomography (PET) in control and Wisket rats. No significant differences were identified in the basal whole-brain metabolism between the two groups, and the metabolism was not modified acutely by a single intraperitoneal caffeine (20 mg/kg) injection in either group. However, one day after caffeine administration, significantly enhanced ^18^F-FDG uptake was detected in the whole brain and the investigated areas (hippocampus, striatum, thalamus, and hypothalamus) in the control group. Although the Wisket animals showed only moderate enhancements in the ^18^F-FDG uptake, significantly lower brain metabolism was observed in this group than in the caffeine-treated control group. This study highlights that the basal brain metabolism of Wisket animals was similar to control rats, and that was not influenced acutely by single caffeine treatment at the whole-brain level. Nevertheless, the distinct delayed responsiveness to this psychostimulant in Wisket model rats suggests impaired control of the cerebral metabolism.

## 1. Introduction

Schizophrenia is a devastating neuropsychiatric disease with very complex symptoms and signs including positive ones, such as delusion and hallucination, negative ones, for example, lack of motivation, and cognitive impairments, such as decreased learning and memory capacity. Positron emission tomography (PET) is a non-invasive technique that detects the tissue distribution of specific labeled tracers. Fluorinated glucose analog ^18^fluorodeoxyglucose (^18^F-FDG) is one of the most used tracers in clinical diagnostics, and it can accumulate in metabolically active tissues in an activity-dependent manner.

Conflicting results have been reported on altered metabolism in various brain structures in schizophrenia patients by PET studies [1,2,3,4,5]. While preclinical models cannot represent the full picture of schizophrenia (e.g., most of the positive symptoms of schizophrenia request verbal report to be measured properly), preclinical studies also suggest changes in brain metabolism by using various single-hit animal models of schizophrenia, including maternal immune activation, neonatal hippocampal lesion, the administration of NMDA receptor antagonists, mutant mice of D_2_ dopamine or metabotropic glutamate 5 (mGluR5) receptors, or microtubule-associated protein (STOP: stable tubule only peptide) [6,7,8,9,10,11,12,13].

Since the etiology of schizophrenia involves the interaction of genetic, developmental, and environmental factors, multiple-hit translational models might provide animals with a high level of validities (constructive, face, and predictive) with a wider range of schizophrenia-related signs. Therefore, a triple-hit rat model named Wisket was developed in our laboratory by combining environmental (post-weaning social isolation for 4 weeks), pharmacological (NMDA receptor antagonist, ketamine, daily treatment intraperitoneally: 30 mg/kg at the age of 4 weeks), and genetic (selective breeding based on behavioral phenotype (pain sensitivity, cognitive function, and sensory gating tests) for more than 40 generations) manipulations [14,15]. The Wisket animals showed a wide range of behavioral disturbances including impaired pain sensitivity, sensory gating, and cognition [16,17,18,19]. Alterations in opioid, cannabinoid, oxytocin, and dopamine receptor (D_2_R and D_1_R) signaling and/or expression together with electroencephalography changes were also detected [14,18,20,21,22,23]. Furthermore, a histological study revealed significant decreases in the frontal cortical thickness and the hippocampal area, moderate increases in the lateral ventricles, and cell disarray of the CA3 subfield of the hippocampus of the Wisket animals [24]. Since volumetric analyses cannot reliably predict the changes in the metabolic pattern [2], the first goal of the study was to characterize the metabolic activity in the whole brain and in various brain areas (cingulate cortex, hippocampus, striatum, amygdala, thalamus, and hypothalamus) related to the cognitive functions of control animals and Wisket animals by ^18^F-FDG PET.

Caffeine is the most widely consumed psychostimulant worldwide, and it can act primarily as a non-selective adenosine A_1_/A_2A_ receptor antagonist and may lead to reduced drowsiness and enhanced locomotor activity [25,26]. Besides the disrupted dopaminergic and glutamatergic neurotransmissions, adenosine dysfunction may also contribute to the etiology of schizophrenia [27]. Increased coffee intake is well documented in patients, but the effects seem to be controversial [28,29,30]. Caffeine may evoke psychosis; however, it can also improve negative symptoms and/or compensate for the antipsychotic medication-induced side effects [29,31]. Our recent study demonstrated that caffeine treatment acutely blunted the cognitive impairments in Wisket animals, while it produced delayed behavioral depression in both the control and the Wisket animals [19]. Several human as well as animal studies investigated the acute effects of caffeine administration on the healthy brain metabolism [32,33,34,35,36,37]. Nevertheless, these effects have not yet been investigated in schizophrenic patients or preclinical models. Therefore, our further aims were to reveal the acute and delayed effects of a single caffeine treatment on brain ^18^F-FDG uptake of control and Wisket animals (Figure 1). The time paradigms of the experiments were the same as in our recently conducted behavioral study [19].

## 2. Results

Regarding the body weight, no significant differences were found between the Wistar and Wisket groups during both the acute (435 ± 14.1 vs. 417 ± 16.7 g, respectively) and the delayed (438 ± 11.3 vs. 433 ± 7.8 g, respectively) series.

### 2.1. Whole-Brain Metabolic Activity

Analysis of the whole-brain metabolism 30 min after caffeine administration (Series 1: acute effect, Figure 1) showed no significant effects on either the treatment or the control groups (Figure 2). In contrast to the acute paradigm, factorial analysis of the whole-brain data one day after the caffeine treatment (Series 2: delayed effect) showed significant effects of the treatment (F_(1,20)_ = 12.29; *p* < 0.005) and close to significant effects of treatment and group interactions (F_(1,20)_ = 3.65; *p* = 0.07; Figure 2). The post hoc comparison revealed that the basal activation measured after saline treatment (see section Experimental Paradigm, Series 1: Acute effects, and Figure 1) was similar in the two groups. Caffeine administration resulted in a significant enhancement of glucose utilization in the Wistar group, but only a limited increase was observed in the Wisket group; therefore, the psychostimulant treatment exhibited a delayed and significant difference between the groups.

Since the delayed caffeine effect was significant at the whole-brain level, the metabolic rates of four brain areas were also analyzed in Series 2.

### 2.2. Hippocampus

The factorial analysis of the hippocampal activity showed a significant effect of the group (F_(1,20)_ = 4.59; *p* < 0.05) and a close to significant effect of the treatment (F_(1,20)_ = 3.55; *p* = 0.07; Figure 3a). The post hoc comparison revealed that the basal metabolic activity of this area was similar in the two groups. Increased hippocampal metabolism was observed after caffeine treatment, primarily in the control animals (*p* = 0.08), and thus, a close to significant (*p* = 0.056) difference was observed between the two groups.

### 2.3. Striatum

The factorial analysis of the striatal activity showed significant effects of the group (F_(1,20)_ = 4.90; *p* < 0.05), treatment (F_(1,20)_ = 8.67; *p* < 0.01) and close to significant group and treatment interactions (F_(1,20)_ = 3.95; *p* = 0.061; Figure 3b). The basal SUV values of this area were comparable between the groups. As caffeine administration caused significantly enhanced metabolism only in the Wistar group, a significant difference was detected between the two groups.

### 2.4. Thalamus

The factorial analysis of thalamic metabolism showed significant effects of the group (F_(1,20)_ = 5.35; *p* < 0.05) and treatment (F_(1,20)_ = 11.74; *p* < 0.005), and close to significant group and treatment interactions (F_(1,20)_ = 3.19; *p* = 0.089; Figure 4a). The post hoc comparison disclosed that the baseline metabolism of this area did not differ by group. Caffeine treatment produced significantly augmented ^18^F-FDG uptake in the Wistar animals but not in the Wisket animals, hence a significant difference was detected by group one day after caffeine administration.

### 2.5. Hypothalamus

The factorial analysis of the hypothalamic activity showed significant effects by treatment (F_(1,20)_ = 5.62; *p* < 0.05) and close to significant effects by group (F_(1,20)_ = 4.24; *p* = 0.053; Figure 4b). The basal metabolism of this area was similar in the two groups. Caffeine administration caused significantly enhanced metabolism in the Wistar, but only a moderate increase in the Wisket group; therefore, significant differences emerged between them in this circumstance.

## 3. Discussion

This study investigated the brain metabolism of control and Wisket animals after the administration of saline or caffeine. The main findings of this study are threefold: the basal brain metabolism of Wisket animals corresponds to the control (1); that was not influenced acutely by single caffeine administration (2); however, the Wisket animals showed divergent delayed responses to caffeine administration compared to the control rats (3).

Several studies support the impaired brain metabolism in schizophrenia, which might be related to widespread cerebral dysfunction and neurochemical alterations [1,2,3,38]. ^18^F-FDG PET assessments of patients have reported primarily decreased metabolic rates in the cortical regions (especially in relation to negative symptoms), amygdala, and hypothalamus, while inconsistent results have been obtained in the basal ganglia, thalamus, and hippocampus [1,2,3,39,40]. Patients with predominantly negative symptoms had greater metabolic abnormalities compared to patients with primarily positive symptoms and healthy subjects, suggesting that the degree of impairment may depend on the type of schizophrenia [41].

Using animal models could exclude the limitations of clinical studies that result in conflicting results, such as heterogeneous symptom profile, differences in medication status, and disease history. While our recent morphometric study on Wisket rats proved moderately decreased brain volume, it was not accompanied by altered metabolic brain activity at the whole-brain level and in the investigated cerebral structures [24]. Regarding the earlier preclinical studies of brain metabolism in single-hit schizophrenia models, inconsistent data are available [6,7,8,9,10,11,12,42]. The in vitro autoradiography studies have found increased glucose metabolism in several brain structures (e.g., ventral tegmental area, substantia nigra, and hypothalamus) but not in the hippocampus, some cortical areas or thalamus in STOP protein mutant mice, or after a neonatal hippocampal lesion [9,10]. Regarding the in vivo PET studies, the maternal immune stimulation did not influence the glucose uptake in the total brain, but the glucose uptake was decreased in the ventral hippocampus and prefrontal cortex, whereas it was enhanced in some subcortical nuclei [7,12]. Chronic treatment with an NMDA receptor antagonist caused decreased metabolism in all the detected brain regions (caudate putamen, medial prefrontal cortex, cingulate cortex, and hippocampus) [8]. In contrast, mGluR5 mutant mice showed no differences in the baseline SUV values in the investigated brain areas (cortex, hippocampus, thalamus striatum, cerebellum, and the whole brain) compared to controls [11]. The selective dopamine D_2_ receptor deletion from parvalbumin positive interneurons caused decreased metabolism in the somatosensory/insular cortex and lateral hypothalamic areas, but augmented glucose utilization was detected in the basolateral amygdala [6]. Additionally, post-weaning isolation rearing resulted in a reduced metabolic rate in the hippocampus and thalamus [42]. Thus, parts of these studies have achieved agreement with our results obtained in Wisket animals, whereas the inconsistent results might be due to the differences in the animal models and/or the experimental set up [7,17,43,44]. In summary, our results suggest that the impaired brain structure was not accompanied by altered brain metabolism in this triple-hit schizophrenia-like rat model [24].

Regarding the effects of caffeine on the brain metabolism, it had no significant influence shortly after its administration in either group, while it produced a delayed (24 h later) and significant (or close to significant) increase in the investigated areas in the control groups compared to the baseline values. In addition, only a moderate increase was observed in the Wisket animals, leading to a high level of significant difference between the two groups, suggesting distinct delayed responsiveness to this psychostimulant in Wisket model rats.

It is well accepted that the widespread effects of caffeine are related primarily to the blockade of adenosine receptors (A_1_ and A_2_), while other mechanisms (phosphodiesterase inhibition, calcium mobilization, interaction with benzodiazepine, and/or prostaglandin receptors) are only slightly involved [25]. There is considerable evidence to suggest that adenosine decreases the firing of central neurons; furthermore, the adenosinergic system is linked to motivation and cognitive processes by influencing the dopaminergic, glutamatergic, serotoninergic, and cholinergic neurotransmissions [25,36,45,46,47,48]. The activation of adenosine A_1_ receptors, which are present in almost all brain areas, results in a pronounced decrease in transmitter release in several brain areas related to behavioral control [25]. Adenosine A_2_ receptors are concentrated in the dopamine-rich regions of the brain (striatum, nucleus accumbens, and tuberculum olfactorium) but they can also be detected in the hippocampus and cortex. Although most of the schizophrenia symptoms are primarily due to disturbed dopaminergic and glutamatergic neurotransmissions, alterations in the adenosinergic systems have also been shown [27,47]. Adenosine A_1_ and A_2A_ receptor agonists reverse both hyperdopaminergia and NMDA receptor hypofunction-related symptoms, while caffeine may worsen the positive symptoms of schizophrenia patients [29]. However, the data also suggest that caffeine may produce procognitive effects by inhibiting these receptors, even in schizophrenia or in preclinical models [47,49,50,51,52], but the results are inconsistent [53,54,55,56].

The effects of caffeine on cerebral metabolic activity have been primarily investigated with the help of functional magnetic resonance imaging techniques in human participants with controversial results [33,57,58,59], which might be due to the different techniques applied, the dietary caffeine consumption pattern, and/or the caffeine abstinence periods. To understand the effect of caffeine on brain metabolic profiles, caffeine-naive preclinical animal models could be more suitable. The in vitro autoradiography rodent studies have found caffeine-induced increases in the energy metabolism in several brain structures, including limbic and cortical areas, which may reflect its general stimulatory role and positive effects on alertness [25,34,35,36]. Only one PET study has investigated the dose-dependent acute cerebral metabolic responses to caffeine 10 min after ^18^F-FDG injection for 60 min [37]. While the lower dose of caffeine (2.5 mg/kg) had limited effects on the brain metabolism in any investigated brain structures; enhanced metabolism was detected in the striatum, hippocampus, and thalamus, but not in the whole brain, amygdala, or hypothalamus, after 40 mg/kg dose of caffeine. The differences between the results of the above-mentioned study and our data might be due to, at least partially, the differences in the applied dose of caffeine and the experimental paradigm. Based on our present findings, the acute enhancement of locomotor and exploratory activities in control rats and the improved behavior of the Wisket rats did not depend on enhanced whole-brain energy metabolism in response to caffeine [19]. As caffeine causes acute vasoconstriction leading to reduced cerebral blood flow, it might have blunted the direct metabolic effects of adenosine receptor inhibition in in vivo circumstances at the total brain level [60].

Regarding the delayed effects of caffeine, in spite of the significantly decreased motor activity and learning ability in both groups one day after caffeine administration, caffeine caused significant enhancement in brain metabolism only in the control animals, suggesting perturbed delayed metabolic responses to this psychostimulant in our model rats [19]. It is very difficult to explain the controversies between the delayed effects of caffeine on the behavioral activity (decreased) and glucose utilization (enhanced) of the control animals. Since even the withdrawal from chronic caffeine consumption did not cause persistent changes in the availability of A_1_ receptors, we may exclude that the enhanced brain metabolism was due to changes in their number [61]. However, enhanced binding potency of adenosine receptors was observed after the cessation of repeated caffeine treatment; therefore, an enhancement in the A_2B_ activation might stimulate the glucose uptake [62,63]. Another possible explanation might be that the acute vasoconstrictor effects of caffeine disappeared the next day, and thus the activation of the brain structures became uncovered. Furthermore, astrocytes also contribute significantly to the ^18^F-FDG signal [64]; therefore, the delayed enhancement in the glucose uptake might be due to the sum of the enhanced uptake by different cells in the brain after the relief of the vasoconstriction, in spite of the behavioral depression. The effects of anesthesia and/or interactions with caffeine should also be considered [37]. However, further studies are required to characterize the delayed effects of psychostimulants at behavioral and cellular levels. Caffeine-induced increases in dopamine levels in several reward centers may have increased neuronal activity, as supported by findings from a previous study using autoradiography [36]. Reduced dopamine D_2_ receptor function detected in the Wisket animals might also be associated with a reduced metabolic response to caffeine [22,65]. Therefore, the diminished D_2_ receptor function of Wistar rats may result in poor metabolic adaptation, as it has also been reported in schizophrenia patients [66]. In summary, the delayed stimulatory effects of caffeine on the brain metabolism in the control group might be due to changes in the adenosine receptor functions at neural, glial and/or vascular levels, which were disturbed in the Wisket animals.

Some limitations should be considered in the interpretation of our results. The lack of precise anatomical localization with PET is a recognized problem [41]. It is difficult to determine detailed structures for the image; therefore, in the present study, we outlined ROIs based on 3D coordinates for an atlas of the rat brain, but our lack of magnetic resonance imaging for brain structures still poses a limit to this study. For that reason, to minimize the error of the evaluations, only data from relatively large structures were involved in this study.

Furthermore, the group size seems small, but several studies used similar number of animals [37,42,67], and in both series of experiments the saline-treated control and Wisket animals showed similar values, suggesting a good reliability of the obtained parameters.

Importantly, in agreement with the 3Rs (Replacement, Reduction, and Reinforcement) of animal research, we did not involve saline-treated control groups, since PET had a good level of test–retest stability, and the order of different interventions did not influence the brain glucose uptake [6,8,11,68,69].

## 4. Materials and Methods

### 4.1. Experimental Paradigm

Male Wistar (control) and Wisket rats aged between 4 and 5 months were involved in the study. The Hungarian Ethics Committee for Animal Research (RN: XIV/1248/2018 in accordance with EU Directive 2010/63EU) approved the experiments. The animals were group-housed, 3 animals per cage (except during the experimental days, when they were kept individually up to waking after PET scanning), and kept with a 12 h light/dark cycle under controlled temperature (22 ± 1 °C) and humidity (55 ± 10%).

Standard semi-synthetic diet SDS VRF-1 (Animalab Ltd., Vác, Hungary) and water were available ad libitum, except during the experiments. Based on earlier studies, PET scans were carried out on each animal twice with 14 days between the experiments, whereby each animal acted as its own control [6,8,11,69]. The rats were injected with 11.3 ± 0.8 MBq of ^18^F-FDG in 100 μL saline via the lateral tail vein and/or saline or caffeine (20 mg/kg; 4 mL/kg intraperitoneally; Sigma-Aldrich Ltd., Budapest, Hungary). For the injections the animals were anesthetized for a few minutes with 3% Forane using a dedicated small animal anesthesia device. After arousal (within 5 min) the animals were free to move in their cage without behavioral analysis until the beginning of the PET scan. The experiments were performed between 8:00 AM and 12:00 AM to exclude the diurnal variation of brain metabolism. Laboratory animals were kept and treated in compliance with all applicable sections of the Hungarian Laws and regulations of the European Union (permission numbers: III/6-KÁT/2015; 10/2019/DEMÁB).

Series 1 (Acute effects)

In this series (*n* = 5/group), on Week 1, the intraperitoneal saline and intravenous tracer injections were administered consecutively (Figure 1). During the 30 min period of tracer uptake, rats were unrestrained and free to move in their cage, and then the animals were scanned for 30 min, which served as a baseline (see below). Two weeks later, the paradigm was repeated with an intraperitoneal caffeine injection (instead of saline) to determine its acute effect on the metabolic activity. In this trial, a 30 min delay was applied for the development of the caffeine effects and tracer uptake, since this time delay caused significant stimulation in behavioral activity in our recent study [19].

Series 2 (Delayed effects)

In this series (*n* = 6 rats/group), the intraperitoneal saline (on Week 1) and caffeine (on Week 3) treatments were applied one day before the tracer administration and the scanning procedure (Figure 1). The time paradigm in this series is the same as in our recent behavioral study [19].

### 4.2. In Vivo PET Imaging

Thirty minutes after the intravenous radiotracer injection, PET scans were performed in the prone position under isoflurane anesthesia using the preclinical *MiniPET-II* device (Debrecen, Hungary) [70,71]. The body temperature of the animals was controlled by a heating pad (ATC 1000, World Precision Instruments, Rancho Cordova, CA, USA), and they were returned to their home cage, monitored until awake, and given food and water.

Scanner normalization and random correction were applied to the data, and the images were reconstructed with the standard expectation–maximization (EM) iterative algorithm. The voxel size was 0.5 × 0.5 × 0.5 mm, and the spatial resolution varied between 1.4 and 2.1 mm from central to 25 mm radial distances. After 3D OSEM-LOR image reconstruction, volumes of interest (VOIs) were drawn around the examined regions in the transaxial view using the BrainCAD image analysis software. The regions were selected according to the Waxholm Rat atlas (https://scalablebrainatlas.incf.org/rat/PLCJB14; accessed on 1 July 2022). The quantitative ^18^F-FDG PET accumulation was expressed in standardized uptake values (SUVs) using the following formula: SUV = [ROI activity (MBq/mL)]/[injected activity (MBq)/animal weight (g)] (ROI: region of interest). SUV mean values were calculated for the whole brain, hippocampus, striatum, thalamus, and hypothalamus.

### 4.3. Statistical Analysis

All data were expressed as means ± S.E.M., and significance was accepted at the *p* < 0.05 level. Factorial variance analysis was performed to determine the significance level of group (control or Wisket) and treatment (saline or caffeine) in both series and in the SUV values for all the investigated brain regions. When the global test was significant, the Fisher LSD post hoc test was used for the evaluation of the effects of various factors. For the statistical analyses, the STATISTICA 13.5.0.14 (TIBCO Software Inc., Palo Alto, CA, USA) software was used.

## 5. Conclusions

To our knowledge, this study is the first one to show functional changes within the brain in a multiple-hit rat model of schizophrenia. The basal brain metabolism and the acute and delayed effects of caffeine treatment in Wisket and control (Wistar) animals were characterized. The schizophrenia model rats did not show impairment in basal cerebral metabolism. Single caffeine treatment resulted in a delayed increase in cerebral energy metabolism in different regions of the control rats, whereas a blunted response could be observed in the Wisket group. Our study also suggests that neuroimaging may help to elucidate the nature of the effects of different interventions in the cerebral metabolism in this type of neuropsychiatric disease model.

## Figures and Tables

**Figure 1 ijms-23-08186-f001:**
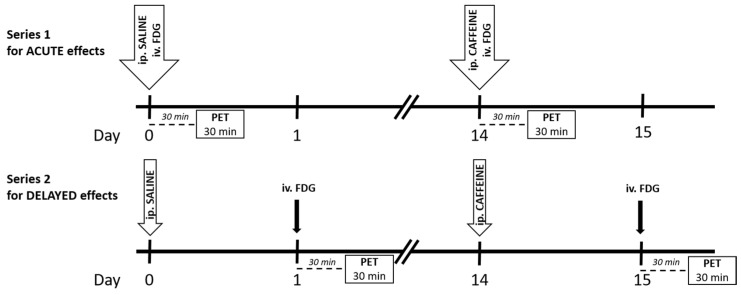
Time scale of the experimental paradigm for both series. Abbreviation: FDG: ^18^F-FDG. The tracer was administered intravenously (iv.), and saline and caffeine were injected intraperitoneally (ip.). The PET was performed 30 min after tracer injection for 30 min.

**Figure 2 ijms-23-08186-f002:**
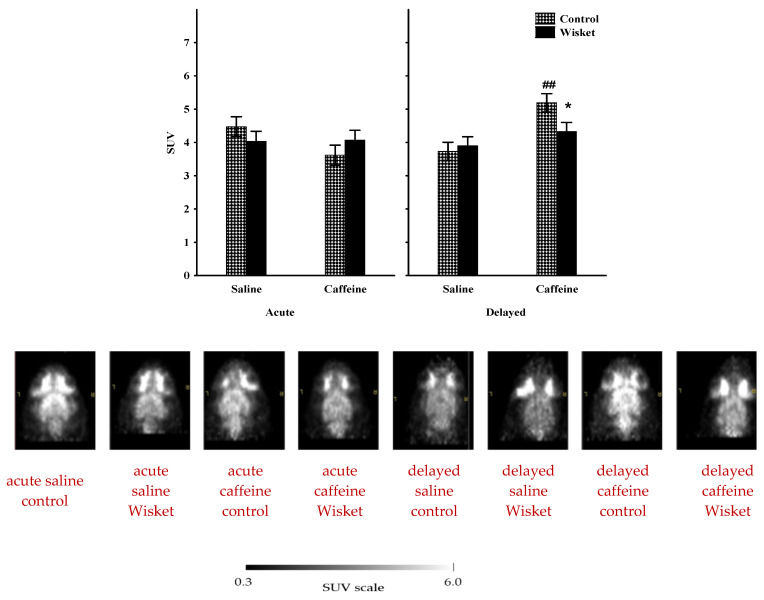
The mean ^18^F-FDG PET SUV with standard error bars (with their representative horizontal PET images and SUV scale) in the whole brain of control and Wisket animals 30 min or one day after saline or caffeine administration. Symbols denote significant differences between the two groups (*) and compared to saline (##); *p* < 0.05 or *p* < 0.01 with one or two symbols, respectively.

**Figure 3 ijms-23-08186-f003:**
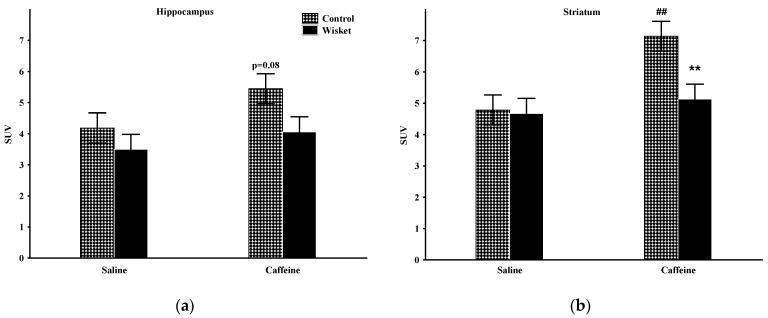
The mean ^18^F-FDG PET SUV with standard error bars in the hippocampus (**a**) and striatum (**b**) in control and Wisket animals with saline/caffeine applied one day before PET scan. Symbols denote significant differences between the two groups (**) compared to saline (##); *p* < 0.01 with two symbols.

**Figure 4 ijms-23-08186-f004:**
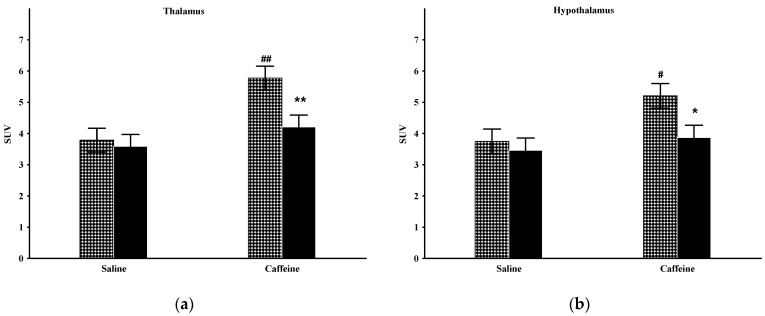
The mean ^18^F-FDG PET SUV with standard error bars in the thalamus (**a**) and hypothalamus (**b**) of control and Wisket animals with saline/caffeine applied one day before PET scan. Symbols denote significant differences between the two groups (* or **) compared to saline (# or ##); *p* < 0.05 or *p* < 0.01 with one or two symbols, respectively.

## Data Availability

Data are available on request.

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
