# Peer review of "Caffeine-Induced Acute and Delayed Responses in Cerebral Metabolism of Control and Schizophrenia-like Wisket Rats"

_ijms, 2022, doi:10.3390/ijms23158186_

Round 1

Reviewer 1 Report

Autors evaluated the effect of caffeine on acute and delayed cerebral metabolism responses bei PET in different brain regions of Wisket rats as model of Schizophrenia.

The study is well structured but should be improved as follows:

- the group size is quite small. If possible, more animals should be evaluated

- Fig. 2 It should be clearly labeled which image belongs to which group

-Discussion: lane 237-238: this sentence is too far catched as authors only evaluated an in preclinical model and no patients. Please rephrase.

-Discussion: lane 239-244: Comparing the here presented data with effects of a completely different drug AND in a completely different model is also to far catched. Please delete chapter.

-Discussion is very long and should be reduced. Compare data only with relevant studies, that will also reduce the length of the reference list.

-Method: Lane 313-315: Sentence is very confusing and content should be specified

-Method: Authors should consider to evaluate the effect of acute caffein slightly later (1h) as it is shown that caffeins peaks later in the blood after (oral) administration as 30 min.

-whole study needs to be improved for English language. Recommend editing by native speaker.

Reviewer 2 Report

The manuscript by Horvath et al., was a relatively comprehensive research article on the potential roles of brain metabolism changes in the brain disorders of the rat. The author tested effects of caffeine for the rat brains and focused on the regional differences. There were some moderate concerns:

(1) For symptoms (such as Page 1 Lines 32-33), please indicate whether it could be studied using animal models.

(2) Rationale of selecting time points was unclear in Figure 1 and related design.

(3) Statistical analysis such as in Page 4 Lines 111-112, needed power analyses.

(4) The revision needs to address how to scale the regional differences in the animal.

(5) Missing cellular and molecular interpretation for the neural and vascular mechanisms of the treated rats.

(6) Timing of the measurement involves a period of non-active status (in Page 8 Line 295). Please justify the experimental design and behavioral observations. 
